# Augmenting Screw Technique to Prevent TLIF Cage Subsidence: A Biomechanical In Vitro Study

**DOI:** 10.3390/bioengineering12040337

**Published:** 2025-03-24

**Authors:** Alina Jacob, Alicia Feist, Ivan Zderic, Boyko Gueorguiev, Jan Caspar, Christian R. Wirtz, Geoff Richards, Markus Loibl, Daniel Haschtmann, Tamas F. Fekete

**Affiliations:** 1Department of Neurosurgery, University of Ulm, 89081 Ulm, Germany; 2AO Research Institute Davos, 7270 Davos, Switzerland; 3Department of Spine Surgery, Schulthess Clinic, 8008 Zurich, Switzerland

**Keywords:** subsidence, lumbar spine surgery, osteoporosis, biomechanics, TLIF

## Abstract

(1) Cage subsidence in spine surgery is a frequent clinical challenge. This study aimed to assess a novel screw augmentation technique for Transforaminal Lumbar Interbody Fusion in cadavers of reduced bone mineral density (BMD). (2) Forty human lumbar vertebrae (BMD 84.2 ± 24.4 mgHA/cm^3^, range 51–119 mgHA/cm^3^) were assigned to two groups: augmenting screw group and control group. The augmentation technique comprised placement of two additional subcortical screws. Ten constructs per group were loaded with a quasi-static load-to-failure protocol and other ten were cyclically loaded. Failure modes were documented. (3) During the quasi-static load-to-failure testing, the augmenting screw technique showed a significantly higher failure load (1426.0 ± 863.6 N) versus the conventional technique in the control group (682.2 ± 174.5 N, *p* = 0.032). Cyclic loading revealed higher number of cycles and corresponding load until reaching 5 mm subsidence and significantly higher number of cycles and corresponding load until reaching 10 mm subsidence for the augmenting screw technique (9645 ± 3050; 1164.5 ± 305.0 N) versus the conventional technique in the control group (5395 ± 2340; 739.5 ± 234.0 N, *p* < 0.05). Failure modes were different and showed bending of the augmenting screws, followed by cut-out. (4) The investigated augmenting screw technique demonstrated higher failure loads and cycles to failure against cage subsidence compared to conventional cage placement. Failure modes were different between the two techniques and may lead to a different kind of complications.

## 1. Introduction

Annually, 39 million individuals worldwide are diagnosed with lumbar spondylolisthesis, 103 million with lumbar spinal stenosis, and 403 million with symptomatic lumbar disc degeneration [1,2,3]; given the demographic trends, a further rise in incident rates is expected [4]. The increase in degenerative diseases of the lumbar spine has led to a worldwide rise in the number of lumbar fusion surgeries, in which intervertebral cages are frequently used [2,5]. The benefits of a correctly placed cage include the increase in disc and foraminal height and the correction of lordosis. Biomechanically, an intervertebral cage absorbs axial compression forces and thereby decreases the load on the screw–rod construct, which potentially reduces the risk of construct and implant failure [6,7]. Transforaminal lumbar interbody fusion (TLIF) surgery is an established treatment option for degenerative diseases of the lumbar spine including foraminal stenosis, osteochondrosis, spinal canal stenosis, and spondylolisthesis [8]. Although TLIF shows various advantages over other types of procedures and achieves decent clinical outcomes with high fusion rates [9], cage subsidence is one of the most common complications in TLIF surgery. The reported rate of cage subsidence in TLIF is up to 51.2% [10]. Subsidence is not necessarily associated with poor outcomes for the patient but can result in loss of correction and foraminal stenosis. If severe subsidence occurs, the biomechanical advantages of cages are lost. Potential long-term complications include construct failure up to kyphotic deformity, implant loosening, and pseudarthrosis [10,11,12]. Risk factors for cage subsidence are low bone mineral density (BMD) [13,14], specifically in the endplates [15], endplate weakening during disc space preparation [16], and cage geometry [11,17,18,19,20,21,22]. Singular cages with a smaller contact area are more likely to subside than multiple and customed cages with a larger contact area due to higher peak loads acting on the endplate [18,22,23]. Endplate resistance to subsidence is strongest in the posterolateral region, and weakest in the central and anterior regions [24]. The typical cage geometry and preferred placement for optimal correction of the sagittal profile make TLIF cages susceptible to subsidence. TLIF cages are typically singular cages and have a significantly smaller footprint coverage than anterior lumbar interbody fusion and lateral lumbar interbody fusion cages due to limited access to the disc space [25,26]. Their placement is often preferred in the anterior third of the vertebra for lordosis restoration [10,25].

In low bone quality, sinking of a cage which cannot be halted may occur. Cases have been observed where the cage eventually settles on a pedicle screw. At this stage, alignment correction is lost, and the construct has failed. This observation raises the question of whether targeted subcortical screw placement could help prevent cage subsidence at an earlier stage. Therefore, the present study aimed to biomechanically assess whether bicortical screw augmentation can mitigate the subsidence of TLIF cages in specimens of reduced BMD. To the best of our knowledge, this technique has not been evaluated biomechanically for TLIF cages so far.

## 2. Materials and Methods

The Null hypotheses tested were that (1) the maximum failure load of TLIF cage subsidence cannot be altered by subcortically placed augmenting screws, (2) the number of cycles to failure cannot be increased by augmenting screws, and (3) failure modes are indifferent to TLIF cages without augmentation.

### 2.1. Specimens and Preparation

Forty fresh-frozen human cadaveric lumbar vertebrae (two L1, twelve L2, twelve L3, six L4, eight L5) from four male and four female donors aged 82 years on average (range 70–94 years) were used. Quantitative computed tomography (CT) scanning was performed (Revolution Evo, GE Healthcare Chicago, IL, USA) to measure the trabecular BMD using visualization software (AMIRA, Visage Imaging, Berlin, Germany). Trabecular BMD was 77.9 ± 34.3 mgHA/cm^3^ (mean ± standard deviation, SD). Based on BMD, the vertebrae were assigned to two clusters of twenty vertebrae each for treatment with a TLIF cage with or without additional subchondral screw augmentation. Within each cluster, ten constructs (*n* = 10) were further assigned for either destructive quasi-static or cyclic biomechanical loading.

### 2.2. Surgical Technique

Prior to preparation and biomechanical testing, all specimens were thawed at room temperature overnight, the soft tissue was carefully removed, and the vertebrae were isolated. The intervertebral discs were removed on both sides, with special care taken to preserve the integrity of the endplates.

Figure 1 schematically illustrates the surgical technique. The augmentation technique commenced with the removal of the dorsal structure using an oscillating saw to avoid compression force transmission through the posterior structures.

The isolated vertebral body was then placed upside down on a custom-made 3D-printed drill guide which ensured uniform screw placement 4 mm below the endplate. The entry points were set at the left posterolateral third of the vertebra in the area that is accessible during a common approach during in vivo TLIF surgery. The bony bridge between both entry points was maintained at a minimum of 5 mm to prevent the creation of a potential weak point for cortical fracture between the screws. The screw trajectories were not parallel and aimed to ensure optimal support of the TLIF cage. Following pre-drilling of the holes using a 2.7 mm drill, fully threaded, cancellous 4.0 mm screws (Johnson & Johnson MedTech, Zuchwil, Switzerland) were placed manually. The length of each screw was determined to provide bicortical anchorage. In this context, bicortical anchorage refers to the fixation of each augmenting screw to penetrate both lateral cortices of the vertebral body. The length of the screws ranged between 42 mm and 50 mm, and they were angled relative to each other within a range of 11° to 17°. Correct placement was radiographically verified.

A polyether ether ketone (PEEK) T-PAL cage (12 × 13 × 30 mm, Johnson & Johnson MedTech, MA, USA) was placed in the anterior third of the vertebra in the median sagittal plane.

### 2.3. Biomechanical Testing

Biomechanical testing was performed on a biaxial servo-hydraulic material testing machine (MTS 858 Mini Bionix II, MTS Systems, Eden Prairie, MN, USA) equipped with a 5 kN load cell (MCS10-005, HBM, Darmstadt, Germany). The test setup was in accordance with the ASTM F2077-18 test standards for intervertebral body fusion devices [27] and adopted from Calek et al. [16]. The vertebra was mounted and secured on a tiltable device connected to the machine base, which allowed translation. Vertical load applied by the machine actuator with the interconnected load cell was transmitted via a flat cylindrical indenter, measuring 30 mm in diameter. The loading protocol for quasi-static testing comprised a single uniaxial quasi-static ramp in compression at a rate of 2 mm/min, starting at a manually set preload of 20 N. Cyclic testing was performed at 3 Hz applying a physiological loading profile. Whereas the valley load was held constant at 50 N throughout each test, the peak load, starting at 200 N, was monotonically increased cycle by cycle at a rate of 0.1 N/cycle, until the test stop criterion of 10 mm machine displacement with respect to the test begin was reached. A triggered lateral x-ray was shot every 200^th^ cycle. The test ended when the test stop criterion was fulfilled, i.e., when the actuator displacement reached 10 mm compared to the initial preloaded position.

### 2.4. Data Acquisition and Analysis

Machine data of axial displacement and axial load were continuously recorded from the machine transducers at 2 Hz throughout the quasi-static and cyclic tests. The load-to-failure tests were recorded using continuous lateral fluoroscopy. Based on these, and with respect to the actuator position at the test start, machine displacements of 3 mm and 5 mm were considered as failure criteria for cage subsidence from quasi-static loading. The peak load reached before the corresponding criterion had been fulfilled was calculated. With regard to cyclic loading, the failure criteria were set to 3 mm, 5 mm, and 10 mm, and the corresponding numbers of cycles until fulfillment of these criteria were calculated together with the corresponding peak loads. To document subsidence macroscopically, a video camera recorded all tests.

Statistical analysis among the parameters of interest was performed using the Prism software package (Prism; IBM Corp., Armonk, NY, USA). Anderson–Darling and Shapiro–Wilk tests were used to assess the normality of the data distribution for each group separately. Based on a Gaussian distribution for all data sets, unpaired, two-tailed *t*-tests were used to compare the loads and number of cycles until fulfillment of the failure criteria between the two groups. F-test was used to compare variances of the mean failure loads to determine whether the groups’ variability can be assumed equal. The association between BMD and failure load was analyzed using linear regression. The level of significance was set at *p* ≤ 0.05, and a trend towards significance was defined as 0.05 < *p* < 0.075.

## 3. Results

### 3.1. Quasi-Static Testing

The failure load for 3 mm subsidence did not differ significantly between the groups (887.5 ± 577.0 N in augmenting screw group, 891.4 ± 441.2 N in control group, *p* = 0.956, Figure 2, Table 1). The variances of the failure loads for 3 mm subsidence did not differ significantly between the groups (*p* = 0.392). The means of the failure loads reached during severe subsidence differed significantly between the groups (1426.0 ± 863.6 N in augmenting screw group, 682.2 ± 174.5 N in control group, *p* = 0.032). There was no significant association between BMD and failure load in linear regression analysis in either group. The variances of the failure loads for 5 mm subsidence were significantly different between the groups (*p* < 0.001).

### 3.2. Cyclic Testing

In the second testing protocol, the number of cycles to reach 3 mm subsidence in the augmenting screw group was not significantly different from the control group. The number of cycles to reach 5 mm subsidence was higher in the augmenting screw group compared to the control group and showed a statistical trend but did not reach significance (*p* = 0.068). The number of cycles to reach 10 mm subsidence was significantly higher in the augmenting screw group than in the control group (*p* = 0.015; Figure 3, Table 2).

### 3.3. Failure Modes

The failure modes observed during the quasi-static load-to-failure tests were similar to those seen during the cyclic loading protocol and differed between the two groups (Figure 4). In the control group, the failure mode was an endplate fracture at the site of the cage and continuous sinking of the cage into the vertebra reaching moderate and finally severe subsidence. In the augmenting screw group, the cage first settled into the endplate, then the two subcortical screws were bent slightly directly under the cage in the direction of force action with preserved screw anchorage in the cortices. During this phase, the cage was pushed downwards into the cancellous bone. With test progression, a clear cutting-through of the screws not only through the trabecular bone but also at the cortices of the vertebra was observed, allowing further sinking of the cage into the vertebra, reaching severe subsidence. Cut-out was more evident in the vertebrae that underwent quasi-static load-to-failure test than those tested cyclically.

## 4. Discussion

The main findings of the current study are that (1) the failure load to reach severe TLIF cage subsidence can be altered by subcortically placed augmenting screws, (2) the number of cycles to failure can be increased by augmenting screws, and (3) the failure modes are different between the two techniques. These findings indicate that the augmenting screw technique potentially prevents severe subsidence and may help in selected cases where conventional solutions for subsidence prevention are not feasible.

While the effects of mild subsidence may not necessarily be clinically significant, the clinical relevance of severe subsidence is undoubtedly significant from a biomechanical perspective. Mild subsidence occurs frequently and is not necessarily associated with worse outcomes [10,28]. When severe subsidence occurs, biomechanically, the correction is lost, and the cage loses its ability to support the anterior column, and, clinically, severe subsidence has been described to predict TLIF pseudarthrosis and screw loosening [12]. The investigated augmentation technique may help prevent severe subsidence, but not mild or moderate subsidence.

The augmenting screw technique investigated in the present study mitigated severe subsidence, but not mild and moderate subsidence up to 3 mm. A possible explanation is that the screws were placed 4 mm subcortically. This distance was necessary due to the differing endplate concavity depth of the specimens. Osteoporotic, degenerated vertebrae were selected to better reflect the anatomy of patients undergoing TLIF surgery, but degeneration is associated with a change in endplate anatomy with a more variable surface geometry [29]. To allow for standardized screw placement in all specimens without endplate perforation, the entry point was required to suit the vertebra with the greatest concavity, which was, in the present case, set as 4 mm below the endplate, measured at the lateral cortex. Clinically, the prevention of subsidence appears to be particularly relevant for vertebrae with a deep endplate concavity, as previous studies indicate that shallow endplates are associated with a lower subsidence rate [30,31,32]. It remains uncertain whether screws placed closer to the endplate could engage earlier to diminish the moderate subsidence of TLIF cages. Overall, particularly the prevention of severe subsidence seems clinically important. Specifically, for patients with osteoporotic vertebrae and deep endplate concavity, a proposed solution is the placement of screws 4 mm subcortically.

To prevent subsidence, pharmacological osteoporosis therapy is a widely used option in clinics but may be limited by patient-related contraindications and requires time to improve bone density. It can be costly, and there is a high geographical variability regarding the criteria for reimbursability, as shown by the example of teriparatide [33]. Apart from systemic therapy and reducing patient-based risk factors, biomedical strategies include advancements in cage design and biomaterials, as well as cement augmentation [34]. Wilke et al. were among the first to demonstrate that cement augmentation can restore strength and stiffness in osteoporotic vertebral bodies, helping to prevent cage subsidence [23,26,35]. However, preventive vertebroplasty remains controversial due to the rare but potentially severe complications of cement extravasation and embolism [36,37,38,39,40,41]; additionally, an increased fracture risk of the vertebrae adjacent to the augmented ones has been described [14,42,43].

Wang et al. investigated a screw augmentation technique to prevent subsidence in oblique lumbar interbody fusion combined with anterolateral fixation (OLIF). The authors demonstrated that the trajectory, specifically, the position and angle of the vertebral screws was associated with cage subsidence. The authors concluded that inserting screws as close to the endplate as possible and parallel to each other while keeping the cage inside the range of the superior and inferior screws is an optimal implantation strategy [31]. Differences to the present study are the surgical technique and associated cage geometry, with OLIF cages having a larger footprint compared to TLIF T-pal cages. Wang et al. used one screw above and one below, while the present study investigates the effect of two screws augmenting the cage. Although there are distinct differences between the studies, the main findings of the present study are broadly in line with the findings of Wang et al., suggesting screw augmentation as an alternative to mitigate subsidence.

In the present study, the failure modes observed differed between the groups. The two subcortical screws were first bent slightly under the cage, allowing moderate but no severe subsidence. With test progression, a clear cutting through of the screws, not only through the trabecular bone but also at the cortices of the vertebra, was observed, with further sinking of the cage into the vertebra resulting in delayed severe subsidence. A possible explanation may be that the subcortical screws partially transmitted the axial load to the cortical bone accounting for a higher failure load requiring more cycles to reach severe subsidence. In the present study, fully threaded screws were used. In other anatomic areas, the use of partially threaded screws has been shown to decrease the risk of cut-out and could pose an alternative for vertebral use [44]. There was higher data scattering for the augmenting screw technique compared with the control group. This effect may be attributed to the use of osteoporotic specimens exhibiting degenerative changes including subchondral sclerosis. This increased density close to the endplates could provide improved screw fixation resulting in better screw purchase of the investigated technique. Overall, new failure modes need to be explored further as they may lead to a different kind of complications.

This biomechanical study inherits several limitations. First, isolated axial compression force was applied through the cage to the endplate and vertebra, which does not represent the physiological forces acting on the human spine in vivo. The primary biomechanical role of an intervertebral implant is to support the anterior column and transmit compressive forces, as 80% of all compressive forces are reported to be transmitted through the anterior column. Therefore, we considered this simplified model as appropriate. Second, the setup ensured a maximum contact area by carefully preparing the endplate and precise alignment of the upper surface of the cage upon the isolated vertebra, which could be tilted and translated as needed parallel to the stamp. However, in surgical settings, cage placement is far more complex and can be restricted. Peak loads at sites of smaller contact areas between the cage and vertebra may occur, for instance, when a cage is slightly tilted. In clinics, the threshold of endplate fracture may be lower. Third, screw placement to prevent cage subsidence in the present study was only tested for ideal placement conditions. In vitro conditions allowed unlimited access to an isolated vertebra lateral cage for optimal subcortical screw placement. The feasibility, failure rates, and modes of the investigated technique in vivo need to be assessed.

The concept is unique for TLIF and may prove valuable for selected cases with compromised bone quality, contributing to the existing body of literature. A relatively large sample size of forty fresh-frozen human cadaveric lumbar specimens was utilized in the current work to enhance the statistical power. Biomechanical studies with combined motion patterns and extensive cyclic testing should be performed to determine whether migration or loosening of the subcortical screws can occur. In addition, the biomechanical effects of different entry points and screw trajectories need to be investigated.

## 5. Conclusions

Augmenting screws can help prevent severe cage subsidence in vertebrae with low bone quality when conventional solutions are not feasible. However, this technique should be applied with caution as its failure modes differ and require further exploration to understand potential complications.

## Figures and Tables

**Figure 1 bioengineering-12-00337-f001:**
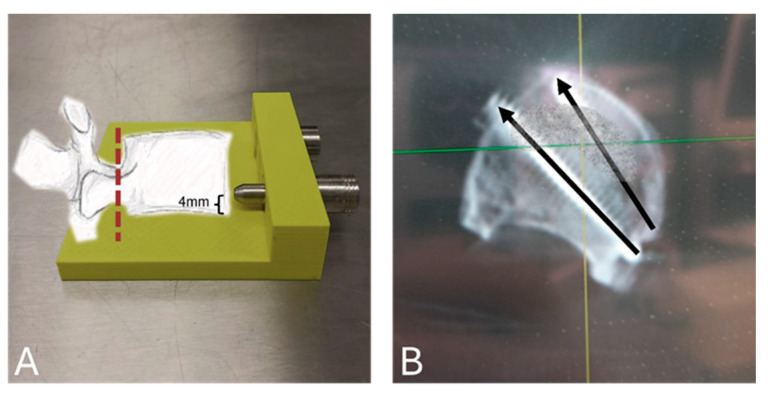
Visualization of the augmentation technique. (**A**) The picture shows a custom-made 3D-printed drill guide which ensured uniform screw placement 4 mm below the endplate. The red dashed line indicates the removal of the dorsal structures. A custom-made drill guide ensured uniform screw placement in all vertebrae 4 mm below the endplate. (**B**) The picture represents a 3D scan of a prepared specimen. The dorsal structures were removed. Two screws were inserted bicortically from the posterolateral third of the vertebra, the area which is accessible during surgery during a TLIF approach.

**Figure 2 bioengineering-12-00337-f002:**
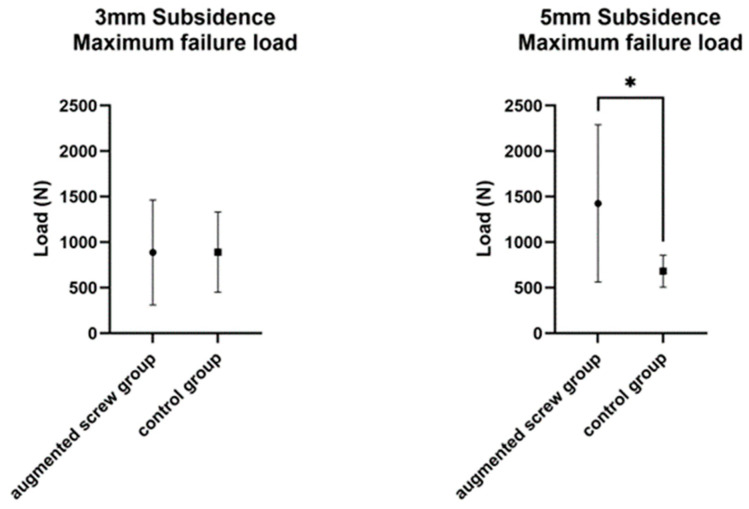
Failure load reached during moderate subsidence (3 mm displacement, **left**) and severe subsidence (5 mm displacement, **right**) of the augmenting screw group and the control group. There was higher data scattering for the augmenting screw technique. Asterisk indicates significance *: *p* < 0.05.

**Figure 3 bioengineering-12-00337-f003:**
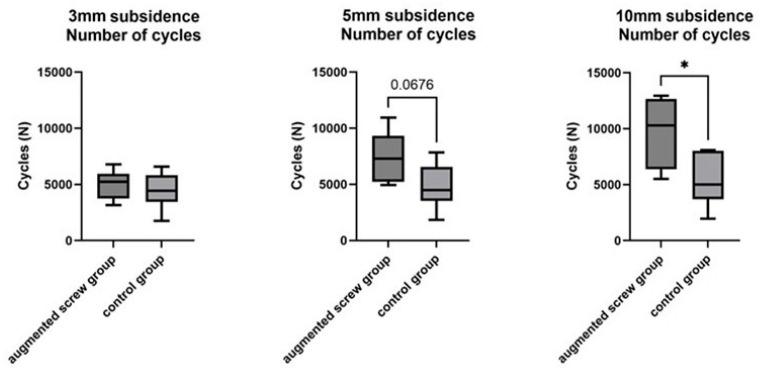
Number of cycles to reach 3 mm subsidence (**left**), 5 mm subsidence (**middle**), and 10 mm subsidence (**right**) of the augmenting screw group and the control group. Asterisk indicates significance *: *p* < 0.05.

**Figure 4 bioengineering-12-00337-f004:**
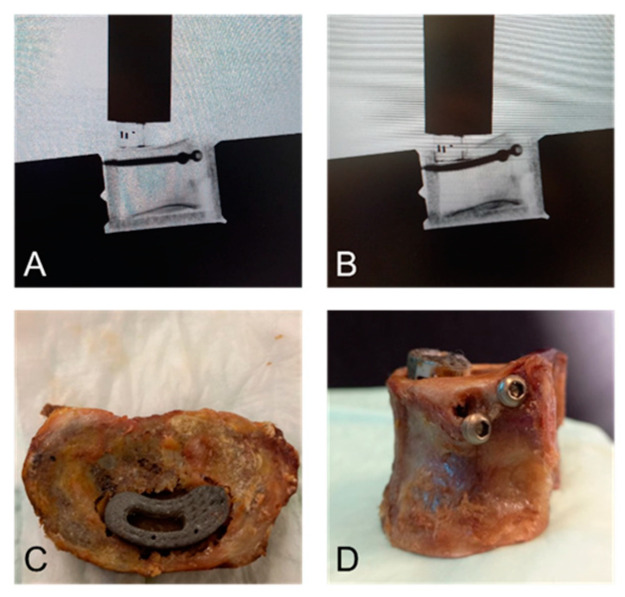
Failure modes. Lateral x-ray of a lumbar vertebra TLIF cage with augmenting screws (**A**) before and (**B**) during quasi-static load-to-failure testing. In the augmenting screw group, bending of the screws was observed radiologically. (**C**) Macroscopically, endplate fracture and cage subsidence were confirmed after axial compression loading, and (**D**) cortical cut-out of the screws was observed.

**Table 1 bioengineering-12-00337-t001:** Results for failure loads.

	Augmenting Screw Group (Mean ± SD) n = 10	Control Group (Mean ± SD) n = 10	*p*-Value	*p*-Value Variance
Failure load 3 mm subsidence	887.5 ± 577.0	891.4 ± 441.2	0.956	0.392
Failure load 5 mm subsidence	1426.0 ± 863.6	682.2 ± 174.5	<0.05 *	<0.001 ***

* Asterisk indicates significance *: *p* < 0.05, ***: *p* < 0.001.

**Table 2 bioengineering-12-00337-t002:** Results cyclic loading protocol.

Variable	Augmenting Screw Group (Mean ± SD) n = 10	Control Group (Mean ± SD) n = 10	*p*-Value
Number of cycles to 3 mm subsidence	4960 ± 1369	4466 ± 1639	0.567
Number of cycles to 5 mm subsidence	7437 ± 2388	4812 ± 2035	0.068
Number of cycles to 10 mm subsidence	9645 ± 3050	5395 ± 2340	<0.05 *

* Asterisk indicates significance *: *p* < 0.05.

## Data Availability

Data available on request due to restrictions eg privacy or ethical.

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
