# Peer review of "Augmenting Screw Technique to Prevent TLIF Cage Subsidence: A Biomechanical In Vitro Study"

_bioengineering, 2025, doi:10.3390/bioengineering12040337_

Round 1
Reviewer 1 Report
Comments and Suggestions for Authors
The authors of this paper present an experimental method to investigate if an improved placement of somatic screws can prevent cage subsidence in TLIF. I am sorry but the first problem that I have with this paper is that the technique described as augmentation is not clear at all and due to the fact that this us an experimental paper, I strongly paper that the figures related to the technique are completely unsatisfactory. The whole procedure should be better detailed so to help the reader its execution in the cadaver, the possibility of reproducing it in the living and the whole sequence of screws trajectory placement ( a short video might work t0o). As a consequence, the discussion is not satisfactory and the authors should spend more time in explaining how this model can be more helpful in improving TLIF execution and why it can be useful only in severe cage loosening and not in moderate degree loosening, considering that the second is much more frequent. FInally, I would ask the authors to discuss what is the role of cage shape in loosening, when compared to the role of screws
Reviewer 2 Report
Comments and Suggestions for Authors
Accessibility for non-specialists could be improved by briefly explaining biomechanical terms (e.g., "bicortical anchorage").
Clarify why TLIF cages are uniquely prone to subsidence compared to other fusion techniques.
Make a stronger link between risk factors (e.g., endplate concavity) and the proposed solution.
The 3D-printed drill guide’s design and screw trajectory specifics (e.g., angulation relative to the cage) are underdescribed.
Include diagrams or explicit numerical ranges for screw angles.
Nonexistent "Figure 3." Ensure consistency in figure numbering and labeling.
The high variance in the augmented group’s failure load (SD ±863.6N vs. ±174.5N control) is attributed to osteoporotic specimens but lacks exploration of confounding factors (e.g., variations in screw trajectory, residual endplate strength).
Linear regression between BMD and failure load is mentioned but not reported, omitting a potential key insight.
Discuss implications of data variability (e.g., clinical reliability of the technique).
Clarify whether cyclic testing peak loads correlate with real-world physiological loads.
